# Recent Advances and Future Perspectives in the E-Nose Technologies Addressed to the Wine Industry

**DOI:** 10.3390/s24072293

**Published:** 2024-04-04

**Authors:** Gianmarco Alfieri, Margherita Modesti, Riccardo Riggi, Andrea Bellincontro

**Affiliations:** Department for Innovation in Biological, Agro-Food and Forest Systems, University of Tuscia, Via S. Camillo de Lellis, 01100 Viterbo, Italy; gian.alfieri@unitus.it (G.A.); margherita.modesti@unitus.it (M.M.); riccardo.riggi@unitus.it (R.R.)

**Keywords:** E-nose, wine industry, volatile organic compounds (VOCs), sensors, flavor analysis

## Abstract

Electronic nose devices stand out as pioneering innovations in contemporary technological research, addressing the arduous challenge of replicating the complex sense of smell found in humans. Currently, sensor instruments find application in a variety of fields, including environmental, (bio)medical, food, pharmaceutical, and materials production. Particularly the latter, has seen a significant increase in the adoption of technological tools to assess food quality, gradually supplanting human panelists and thus reshaping the entire quality control paradigm in the sector. This process is happening even more rapidly in the world of wine, where olfactory sensory analysis has always played a central role in attributing certain qualities to a wine. In this review, conducted using sources such as PubMed, Science Direct, and Web of Science, we examined papers published between January 2015 and January 2024. The aim was to explore prevailing trends in the use of human panels and sensory tools (such as the E-nose) in the wine industry. The focus was on the evaluation of wine quality attributes by paying specific attention to geographical origin, sensory defects, and monitoring of production trends. Analyzed results show that the application of E-nose-type sensors performs satisfactorily in that trajectory. Nevertheless, the integration of this type of analysis with more classical methods, such as the trained sensory panel test and with the application of destructive instrument volatile compound (VOC) detection (e.g., gas chromatography), still seems necessary to better explore and investigate the aromatic characteristics of wines.

## 1. Introduction

The importance of human smell is often underestimated, but recent research highlights its significance in various aspects of human life. Contrary to the belief that humans have a poor sense of smell, studies show that humans have excellent olfactory abilities, being capable of detecting and discriminating an extraordinary range of odors. The human olfactory bulb is quite large and contains a similar number of neurons to that of other mammals, indicating the significance of human olfaction [1].

The first process of the human olfactory system is to breathe or to sniff a smell into the nose. The difference between normal breath and sniffing depends on the quantity of odorant molecules that flow into the upper part of the nose [2]. 

When olfactory receptors capture an odorant, the chemical reactions between them trigger electrical signals as an output. The signals are then transmitted through the glomeruli to the olfactory bulb, where mitral cells and interneurons are located [3]. 

From these electrical outputs, the brain is able to extract different types of information, including the identity of objects and food, environmental hazards, and social and emotional information. This information is probably given by distinct cortical networks within the olfactory system; however, the exact arrangement of these functional networks is not fully understood [4]. The human sense of smell serves a variety of important functions in everyday life [1,4,5]; this is the reason behind the growing interest in the development and application of instruments that mimic our noses [6]. 

Indeed, the application of these types of sensors is now widespread, spanning various fields, from biomedicine [7,8,9,10] and environmental science [11,12,13] to industrial materials production [14,15] and the food industry [16,17,18,19,20,21].

Within the food industry, the wine industry is certainly one of the sectors most interested in developing human sense-like sensors [22]. Hence, wine evaluation with human sense is an old story. For instance, the figure of sommeliers has recovered a significant role in the world of oenology, and since time immemorial, one of the most valuable tools at their disposal has been to train their sense of smell. Moreover, wine sensory analysis includes techniques that go far beyond simple tasting; it is a science that involves the ability to discern the most subtle bouquets and aromas that a wine has to offer [23,24].

A trained nose thus becomes a delicate tool for perceiving complex aromas and translating them to objective evaluations [25]. However, sensory analysis can be imprecise and unreproducible. Furthermore, it requires time, trained panels, and a certified laboratory, and it is therefore not applicable in all contexts. In the wine industry, there is a growing interest in facilitating a technological transition toward the use of new analytical approaches, such as sensor-based methods. In this context, the use of non-destructive devices, such as NIR spectroscopy and E-nose, is becoming more and more popular not only for laboratory use but also for other purposes.

On the other hand, devices such as the E-nose, being fast and easy to use, can supplant possible human errors and be applied at every different stage of production, not only at the final tasting one. 

In this scenario, the review here aims to explore the prevailing trends related to the utilization of the E-nose for volatile organic (VOC) determination and the ascription of aromatic attributes in the field of oenology. To provide a more comprehensive understanding of the subject, a preliminary analysis of the state-of-the-art E-nose concept and a concise overview of recent literature highlighting the primary applications of E-nose technology in the oenological sphere has been performed.

## 2. Review Methodology

Several electronic bibliographic databases (e.g., Web of Science, Science Direct, and PubMed) have been consulted in order to achieve better coverage of the relevant papers published between January 2015 and January 2024. Initially, the review articles were individuated and focused in order to critically select the main documents recently published. Then, starting from the documents preliminarily selected, older literature sources helpful to improve and widen the topic description were included, reaching a total number of 246 papers. Four investigators independently evaluated the available papers by means of predefined eligibility criteria, resolving any disagreement by discussion. The first inclusion criterion was represented by the relevance of articles for discussion focusing on the application of E-noses in the wine industry and the use of panel tests and/or E-nose or GC-MS/E-nose to evaluate the wine quality. In the case of papers dealing with the effect of different factors (i.e., VOCs, quality of wines and grapes, origin of the wine, winemaking process panel test, wine aging, olfactory characterization, E-nose and chemometrics, etc.), we utilized hierarchic approaches to opt for the fitting sections of discussion. In the end, a total of 154 papers were selected and cited. “Mendeley reference manager 1.19.5” software was used for reference management.

## 3. Olfactory Evaluation of Wine: Wet Chemistry and Sensory Analysis

Numerous classes of VOCs have been identified in grapes. Among these, the principal varietal compounds encompass terpenes (predominantly found in grape skin) [26], methoxypyrazines [27], carotenoid-derived compounds also recognized as norisoprenoids [28], thiols [29], benzene derivatives [30], and compounds arising from lipid oxidation [31]. The synthesis of secondary metabolites in grapes—and thus their influence on the chemical composition of the wine aroma—exhibits significant variability, markedly influenced by the concept of “terroir” [32] and agricultural practices [33]. In recent years, numerous researchers have delved into studying the interplay between the vine, soil, and climate on VOCs in grapes and wine [33,34]. Despite these efforts, attempts over the past decade to correlate VOC composition with the wine aroma, as perceived, have often yielded results that are either weak or challenging to interpret [35,36]. As such, the sensory analysis of wine plays a crucial role in the wine industry, enabling the determination of perceptible differences among wines, their characterization, and relevance to consumers. 

The key sensory attributes employed in wine analysis include physiological, psychological, and physicochemical aspects, which are essential for evaluating wine quality and consumer preferences. The aromatic analysis of wine through panel tests involves subjective sensory evaluations by a group of trained individuals [37]. The evaluation by a trained panel and standardized methods ensures the objectivity of professional wine tasting by making it free of personal bias, which is essential for correctly characterizing wine sensory attributes. Moreover, sensory evaluation could transcend the limitations of chemical analysis. It provides insights into how products are perceived beyond their mere chemical composition. By considering factors like aroma, taste, texture, and appearance, sensory evaluation uncovers the nuances that drive consumer preferences and perceptions. 

As a result, the majority of certifications that instruct on wine description within their tasting protocols employ standardized descriptors for various sensory parameters, including mouthfeel, color, and aroma. This practice is prevalent across well-established institutions, such as the Master of Wine Institute (MW), Wine and Spirit Education Trust (WSET3 and dipWSET) [38], the Master Sommelier (MS) certification [39] and its associated materials [40], as well as the certifications provided by the Society of Wine Educators (CSW and CWE) [41,42]. 

While panel tests are valuable tools in assessing wine aromas, several issues and challenges are associated with this approach. One of the main limitations is the inter-panel variability. Hence, different panels may yield varied assessments, as panelists may interpret and describe aromas differently. This inter-panel variability can pose challenges in achieving standardized and reproducible results [43]. The effectiveness of panel tests is highly dependent on the training and experience of the panelists. Inadequate training or lack of experience may result in less accurate and reliable evaluations [44]. Continuous exposure to aromas during a tasting session can lead to sensory fatigue among panelists, affecting their ability to detect and differentiate aromas accurately over time [45,46]. Moreover, a limited vocabulary can often represent a problem. Describing aromas can be challenging, and panelists may have a limited vocabulary to express their sensory experiences [47]. Lastly, panel tests are more qualitative than quantitative, making it difficult to precisely quantify the concentration of specific aroma compounds in a wine [48]. Organizing and conducting panel tests can be re-source-intensive in terms of both time and cost. Training, maintaining, and coordinating a panel require significant investments [49]. 

The matrix effect, due to the richness in wine aromatic compounds and the presence of high levels of alcohol, which can lead to chemical interference effects, masking of aromas, and interaction between compounds, can also be a limiting point for these analyses.

Opposed to sensory analysis for characterizing the aromatic components of wines are wet gas chromatography (GC) analyses. GC is an analytical technique used to separate, identify, and quantify volatile components present in a wine [50]. This technique exploits the ability of compounds to migrate through a stationary phase within a separating column in response to a carrier gas flow [51]. However, like any destructive analytical method, GC presents its limitations that must be considered in the context of wine analysis.

Volatile Compound Selectivity: GC is highly effective in analyzing volatile compounds, but it may not capture all non-volatile or semi-volatile components in wine. This limitation can lead to an incomplete representation of the wine’s chemical composition [52].Matrix Effects: The complex matrix of wine, including various organic and inorganic components, can influence the separation and detection of compounds in GC [53]. Co-elution of compounds and interference from matrix components may occur, affecting the accuracy and specificity of the results.Quantification Challenges: While GC provides excellent qualitative information, quantifying compounds can be a challenge without the use of appropriate internal or external standards. Variability in detector response can also offer quantification difficulties [54].Need for Complementary Techniques: To achieve a comprehensive understanding of wine composition, GC is often coupled with other analytical methods such as Mass Spectrometry (GC-MS) [55] or Flame Ionization Detection (GC-FID) [56]. This integration adds complexity and may increase the cost of the analysis.Time-Consuming Sample Preparation: The preparation of wine samples for GC analysis involves extraction and concentration steps, which can be time consuming. Delicate handling of samples is essential to prevent changes in composition during preparation [57].Instrumentation and Maintenance: GC instruments require regular maintenance, and the quality of results is contingent on the proper functioning of the equipment [58,59]. The need for skilled personnel and the associated costs of maintenance can be limiting factors [60].Cost Implications: While GC analysis provides valuable insights, the initial investment in equipment and consumables can be relatively high [59]. This cost may be a limiting factor for smaller wineries or research facilities with budget constraints [61].

## 4. Application of Electronic Noses in Oenology: Principles, Use, and Main Issues

Sensory analysis was commonly employed to discern the various aromatic characteristics of numerous food products and beverages (including wine). However, this method often proved to be imprecise, lacking in reproducibility, and highly subjective. It was recognized that human senses, including the sense of smell, can be influenced by both physical and mental conditions, as well as external factors. Consequently, sensory analysis began to be paired and compared with analytical instruments, such as gas chromatography, to mitigate these issues. Nevertheless, utilizing such analytical tools entailed significant time, expertise, and costly equipment. Additionally, this approach frequently necessitates intricate sample preparation procedures, which are time-consuming and not always feasible for the modern wine industry’s need for swift and straightforward quality assessment methods. To address these challenges, in recent years, there has been a notable shift towards utilizing tools that emulate the biological sense of smell, known as E-noses. These devices have emerged as some of the most interesting and new instruments in the wine industry, offering a promising solution to the limitations associated with traditional sensory analysis and analytical techniques.

Typically, owing to the advanced sensing performances of its sensor array, the E-nose has the aptitude to convert volatile compounds found within the wine matrix into detectable electric signals, often in digital form. As a consequence, these signals are thoroughly analyzed, primarily during post-processing, to derive a potentially meaningful pattern relevant to the specific analysis at hand. It is evident, therefore, that the primary component of an E-nose tool, which can be tailored to suit various applications, lies in its sensor array. 

Most E-nose systems rely on Conducting Polymers (CPs), Metal Oxide Semiconductors (MOSs), Metal Oxide Semiconductor Field-Effect Transistors (MOSFETs), and mass-sensitive (such as quartz microbalance) acoustic and optic sensors [61]. 

The composition and functioning of the different sensors are described below.


**Metal-Oxide Semiconductor (MOS) Sensors:**



Composition: Metal-oxide semiconducting film (commonly SnO_2_) on a ceramic substrate or TiO_2_ on a carbon nanotube substrate [62,63,64].Operation: In the presence of air, oxygen is adsorbed on the semiconductor surface, creating an electron-depleted region and high-resistance contacts. Exposure to a reducing gas leads to the release of electrons, causing an exponential change in resistance [65,66,67].



**Conductive Polymer (CP) Sensors:**



Composition: Made of organic aromatic or heteroaromatic materials, such as pyrrole, aniline, thiophene, and acetylene, deposited onto a ceramic substrate with gold-plated electrodes [68,69,70,71,72,73,74].Operation: Conductivity is primarily due to electron movement along the extended π system. Exposure to vapor induces a change in electron density, resulting in a measurable change in conductivity [65,73].



**Mass-based sensors:**



Types: Divided into Bulk Acoustic Wave (BAW) or Quartz Crystal Microbalance (QCM) and Surface Acoustic Wave (SAW) detectors [75,76,77,78,79,80,81].Composition: Piezoelectric material (usually a single quartz crystal) coated with a sorbent membrane [80,82,83,84].Operation: An alternating current generates a resonant wave, and when vapor permeates the sorbent layer, the total mass of the film increases, leading to a proportional change in resonance frequency [73,85,86].


Commonly, MOS sensors are likely the most employed for wine analysis thanks to their cost-effectiveness, reliable performance, minimal drift tendencies, and sensitivity to many volatile compounds. Nonetheless, they exhibit poor selectivity and susceptibility. Regarding CP sensors, they are highly sensitive and resistant to poisoning effects compared to MOS devices. However, they exhibit limited reproducibility. In contrast, MOSFET sensors operate on the variations of electrostatic potential. They are generally robust in different environmental conditions, although precise temperature control is advantageous for accurate data interpretation. Finally, mass-sensitive devices, such as piezoelectric sensors, utilize piezoelectricity to convert mechanical variations caused by ligand mass into changes in resonance frequency. Despite their high selectivity, they demonstrate limited stability to fluctuations in temperature and humidity [61]. The environmental conditions associated with this analysis become very significant, as the volatility of aromatic compounds is strongly dependent on the temperature at which the measurement is carried out. This is the motivation of the exigence to find out the best temperature for analytical detection and, above all, to carry out all measurements at the same temperature.

A key point of the E-nose use is that the data derived from the sensor detections require accurate manipulation and interpretation, which are normally performed under the machine learning and chemometrics principles. Among the most commonly used statistical approaches are Support Vector Machines (SVM), clustering algorithms like k-means, multivariate statistical analysis, Artificial Neural Networks (ANN), and, in some cases, more complex algorithms such as those applied in Deep Learning (DL) computation. These are considered among the most popular when used within E-nose systems [61]. Compared to traditional sensory analysis methods, the E-nose offers several significant advantages. In particular, it stands out for greater precision and reliability, proving to be more time-efficient and less susceptible to environmental factors, as well as examiners’ psychological state and condition [45,46]. Unlike analytical methods like gas chromatography, the Electronic Nose emerges for its user-friendly nature, as it does not require extensive expertise for operation and interpretation [61]; therefore, given its ease of use, Electronic Nose sensors are widely studied and utilized in the wine industry. Especially considering that the aromatic profile of different wines provides a range of information regarding attributes, winemaking methods, geographical origin, wine aging, and potential deviations that wines may exhibit. Table 1 presents various applications of E-noses in the field of enology, along with the sensor arrays used and the corresponding chemometric analyses or traditional comparison methods employed for data interpretation (Table 1). 

The E-noses equipped with MOS sensors are commonly used for detecting wine spoilage. These types of sensors have been tested in detecting volatile compounds associated with after-blotting aromatic wine defects, like trichloroanisole (TCA) and corky off-flavors [94,95,96]. Other uses of E-nose QMB sensors are the rapid detection and non-destructive analysis of aromatic defects released during wine production and aging, such as those caused by *Brettanomyces* spp. [93]. E-noses have also been used for the evaluation of wine and grape quality; they have been tested for discriminating between grape musts at different degrees of ripeness and/or coming from different grape varieties. This can be extremely important, allowing us to decide when to harvest wine grapes that have reached a pronounced aromatic maturity, and this can be achieved with E-noses based on MOS, CP, or QMB [98,99,117], improving the aromatic characteristics of the derived wine. Later, electronic noses with MOS technology or QMB/MOS were also used to identify different grape varieties used to make wines [118,119] or evenly used to discriminate the wines themselves using sensors based on porphyrins [113,121]. Other applications of great interest are related to the opportunity to detect wine defects with E-noses based on MOS, such as the presence or absence of reducing chemical conditions [92] related to the acetic bacteria presence and development [97] or any other process deviations which may occur [91,92]. Wine aging is another stage along the wine production chain at which E-Nose application can be exploited, whether in sparkling wines [88], liqueur wines [89], or any others [88]. There is also great interest in the use of MOS-based E-noses to select different winemaking practices [106,111,112] or assess the activity of added enzymes on the aroma pattern [108]. On the other hand, QMB-based E-noses have been used to monitor carbonic maceration in certain types of wine [108], to detect the development of noble rot (Botrytis cinerea) and its effect on the quality attributes of rotten grapes [110], or to monitor the aromatic differences between refermented wines produced with Charmat or traditional methods [109]. A QMB-based E-Nose combined with volatilome analysis has also been tested with the aim of discriminating sparkling wines obtained under different temperature and yeast conditions [88]. The QMB E-nose can also be used to monitor how vine leaf removal may affect the aromatic compounds of the grape berries [104]. It has been found that the application of these non-destructive tools with MOS sensors seems to be widely studied for its aptitude to compare different wines coming from different vintages, even identifying the wine quality [101,102,103,122]. Some applications aimed at monitoring the varietal characteristics of certain wine grape varieties, such as Syrah and Petit Manseng, have also been studied with MOS and CP sensors, respectively [101,124]. These applications utilize various sensors of different origin, functioning together with chemometric analyses, often in conjunction with wet chemistry techniques [132]. Additionally, MOS-based E-noses are used for the evaluation of differences in volatile profiles of sweet wine [126] authentication and quantitation of red wines [123,125,127,129,131]. An MOS-based E-nose, equipped with gas sensors modified with zeolite Y, demonstrated an enhanced performance in detecting wine aroma compounds [130]. The use of ANN data processing, a computational approach inspired by the functioning of the human brain, designed to process and recognize information, can also be underlined [133]. Overall, Electronic Noses play a crucial role in various aspects of oenology, offering rapid, non-destructive, and cost-effective methods for quality assessment, defect detection, and process optimization in winemaking and grape cultivation. The interest related to the use of portable electronic noses is growing significantly, as they can be an added value during the wine production stages, considering that they can be used directly in the field [95,98].

In this overview, however, it has been recognized that the E-nose is a non-specific sensor-based device whose discriminative ability enables it to perceive a pattern or aroma profile much more like the human nose than GC-MS identification [60]. This leads to the fact that the results of E-nose measurements often differ considerably from those of VOCs detected by traditional analytics, an expected observation considering that, in general, VOCs are derived from a quantitative GC-MS analysis of molecules that are present in the analyzed matrix; whereas in the case of tasting or E-nose, not all VOCs are detected [117]. This is because volatile molecules have different perception thresholds and are subject to chemical interactions, combinations, cover-ups, and synergies that can strongly influence their aromatic perception by the human nose or E-nose [134,135]. It is known that the aroma or smell of grape juice is, in most cases, very similar in many grape varieties due to the absence of fermentation processes and enzymatic or chemical reactions [136], even though, after performing GC-MS analyses, a significant difference in VOCs is appreciated.

## 5. Case Study: E-Nose in Application to Detect Smoke Taint in Wines

Wine, due to its extremely complex matrix and the multiplicity of winemaking operations behind its creation, is often susceptible to the presence of certain undesirable aromatic compounds that make the wine defective/unpleasant to taste. Among these compounds, volatile phenols (VPs) are known to be responsible for off-flavor formation in wine. VPs are chemical molecules that can greatly influence the organoleptic characteristics of the wine, contributing both to its flavor profile and overall taste [137,138]. The presence of VPs in wines can be attributed to two different phenomena: the presence of contaminant yeast *Brettanomyces* and the grapevine’s exposure to smoke (Figure 1). Smoke taint, or smoke contamination, is a phenomenon that occurs when grapes are exposed to smoke from nearby fire incursions or forest fires. Among the smoke-derived VPs, guaiacol, 4-methylguaiacol, syringol, and cresol are certainly the most relevant. These compounds are naturally produced through the thermal degradation of lignin, and they can be absorbed into grapes berries, especially if the exposition occurs between veraison and harvest. Once taken in, smoke-derived VPs are swiftly glycosylated and stored in the skin and pulp of grape berries. Throughout winemaking, many of these glycoconjugates are enzymatically broken down into their active free forms, releasing their undesirable ‘smoky’ aromas, with a significant portion of the glycoconjugate pool persisting in the final wine. Moreover, the hydrolysis of VP glycoconjugates within the mouth has been observed, potentially further influencing the flavor and aftertaste of smoke-affected wines. Smoke tainted wines are characterized by ashy, smoky, and medicinal aromatic notes [139,140]. These compounds have very low thresholds of human perception, and they are therefore not strongly influenced by the matrix effect, as happens for other compounds, making them easy to detect in the final product and thus influencing its quality. Smoke taint is an issue of increasing interest for the wine industry since there are significant difficulties in the production of high-quality wine from smoke-affected berries [141,142]. The interconnection between global warming and the increase in wildfires near vineyards poses a concerning challenge for the wine industry. Climate change, characterized by rising temperatures, changes in precipitation patterns, and extreme weather events, has a significant impact on the distribution and intensity of wildfires in many wine regions worldwide [143,144,145]. Smoke and particles from fires can settle on grapes, influencing the aromatic and flavor profile of the wine [146]. This causes an alteration of the Terroir, which includes climate, soil, and the surrounding environment, a key element in winemaking. Winemakers are striving to adopt adaptation and mitigation strategies to address the impacts of wildfires, but adapting to climate change and managing wildfires are complex challenges that require global commitment [147,148]. The presence and concentration of VPs can vary greatly depending on grape variety, growing practices, winemaking techniques, and wine aging. 

Moreover, the presence of smoke taint can be managed through specific winemaking techniques, such as cold maceration and the use of selected yeasts, to minimize the extraction of these undesirable compounds [137]. Ongoing scientific research and evolving oenological practices are constantly contributing to improving strategies for managing these VPs in the world of wine. To evaluate the presence of smoke-derived VPs, including both free form and their glycoconjugates, wine operators typically rely on sending samples of grapes and/or wines to commercial laboratories or conducting small harvests for sensory analysis. However, the considerable expense linked with laboratory testing may hinder many producers from opting for this analysis, and sensory analysis can be time-intensive, potentially preventing timely actions within the constraints of a vintage. Early identification of smoke taint contamination is essential to intervene early and preserve wine quality. Consequently, there is a pressing need for a swift and cost-effective alternative method to assess smoke contamination [139]. The high presence of these compounds in wines permanently affects their quality, even forcing the wineries to devalue their product and lose out on profit opportunities. In fact, consumers who find themselves drinking a bottle contaminated by high VP values are highly dissatisfied unable to appreciate its true organoleptic characteristics.

Advanced monitoring strategies are crucial when dealing with grapes already contaminated by smoke, allowing for the identification of compromised product batches and working on them in the best possible way to produce a final wine unaffected by these undesirable compounds [139]. In this context, the application of E-noses foresees a flourishing and functional future. Table 2 shows some of the most relevant scientific works on the use of E-noses on smoke-tainted grapes and wine (Table 2).

As previously reported, MOS-based E-noses are the most popular devices. For instance, Summerson et al. (2021) tested a low-cost E-nose to distinguish smoke-tainted and non-smoke-tainted wines. Sensor signals were then used as inputs for machine learning modeling in order to classify the wines based on the type of smoke taint amelioration treatment applied and the smoke taint state of affection [139]. The E-nose integrated with machine learning approaches resulted in a valuable, rapid, and cost-effective technique for assessing smoke taint. Other studies evaluated the potential use of E-noses to assess wines made from grapes exposed to different amounts of smoke. The E-nose measurements were coupled with the analytical concentration of VPs and the consumer’s sensory analysis and then elaborated with machine learning modeling strategies. Again, the obtained model highlights that E-nose could be effectively used by winemakers to assess smoke contamination levels and implement amelioration strategies [149]. Other authors used a low-cost E-nose to assess the status of certain wines and pointed out a machine learning model based on ANNs to discriminate among non-smoked and smoked wine, with or without remediation strategies (i.e., activated charcoal treatment and cleavage enzyme). Similarly, researchers used a low-cost E-nose MOS sensor-based and employed ANN computation to analyze volatile aromatic compounds in smoke-tainted Cabernet Sauvignon wines [153]. Accordingly, with the previous observations, the method offered a rapid and potentially accurate means of detecting smoke taint, even complementing traditional techniques such as GC-MS. Lastly, a mass-based E-nose (quartz microbalance—QMB sensors), coupled with chemometric techniques such as PCA and PLS, has been employed to assess the effectiveness of ozone as a remediation strategy to remove smoke taint from grapes [154]. The use of these E-nose devices could be strongly targeted at companies working in areas prone to fires and where the consequences of climate warming are more influential. The alternative is to exploit it on varieties that contain large amounts of hydroxycinnamic acids, such as Sangiovese, Montepulciano, and Cabernet Sauvignon; these phenols act as precursors for the synthesis of VPs coming from *Brettanomyces* metabolism.

Overall, these studies demonstrate the versatility and efficacy of E-nose technology in detecting and assessing smoke contamination in grapevines and wines by offering rapid, cost-effective, and potentially innovative solutions to this pressing issue faced by winemakers.

## 6. Conclusions

Quality in wine is inextricably linked to its aromatic composition. For hundreds of years, certain sensory analysis methods have been codified and used to give qualitative connotations to the smell of wines and their nature. It is precisely in such a modern context that practical and fast instruments that can take the place of human sensory analysis are now making great strides. The application of E-noses on wine and grapes seems to be almost the perfect match, imitating, in some respects, the human perception, as well as removing the aspects linked to the subjectivity of the individual person. Since the second half of the 20th century, the increased knowledge of this technology has led to the realization of not only functional and high-performance instruments but also low-cost E-noses. This last aspect is basically relevant, as it allows even less structured companies the possibility of purchasing and applying these sensors, thus becoming increasingly competitive in the modern market, which, in turn, is more and more focused on high-quality products. Not to be overlooked is the possibility of having portable E-noses that can be used directly in the field to monitor possible problems, such as pathogen development, or to monitor the aromatic maturity of grapes and, therefore, to identify more accurately the right moment when grapes have a richer aromatic profile. In the future, the increasing development of E-nose devices combined with artificial intelligence and algorithms promises to further revolutionize the wine industry. Thanks to the machine-learning performance and advanced data analysis offered by Artificial Intelligence (AI), E-noses will be able to recognize and interpret an even wider range of volatile compounds, enabling a deeper and more accurate evaluation of wine aromas. Sophisticated algorithms will be able to optimize aroma analysis models, identifying complex correlations between the chemical composition and aroma profile. This will enable producers to further refine winemaking techniques, adapting real-time processes to react to variations in raw materials and environmental conditions.

Ultimately, the combination of E-noses with AI and algorithms, the opportunity of purchasing low-cost E-noses, and the increased knowledge about the materials used to make the instruments themselves represent a significant step towards even more sophisticated, precise, and efficient wine production, paving the way for a new frontier of innovation in the world of oenology.

## Figures and Tables

**Figure 1 sensors-24-02293-f001:**
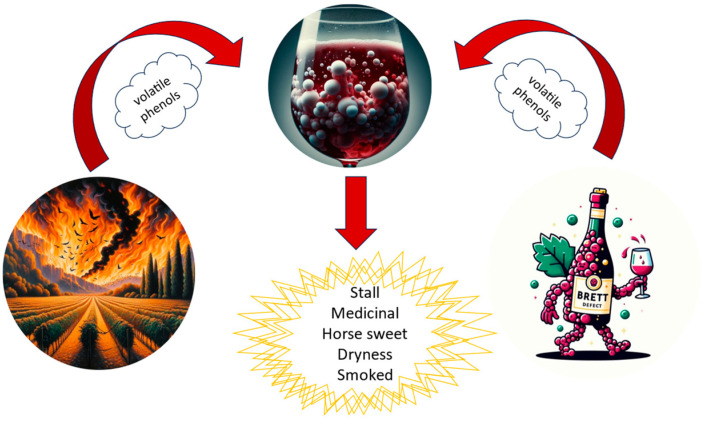
Exposure of grapes to fire or contamination by *Brettanomyces* in grapes and wines can lead to the development of certain defects related to certain volatile phenols.

**Table 1 sensors-24-02293-t001:** Main applications of the electronic nose in the oenological sector.

Category	Application	Sensor Arrays	Chemometrics	Classical Comparison	Reference
**Wine aging**	Evolution of wine over 9 months	MOS	PLS	GC-MS	[87]
	Aging of sparkling wine	QMB	PLS-DA	GC-FID	[88]
	Characterization of refined marc distillates	MOS	PCA	GC-MS	[89]
**Wine defects**	Electronic nose for detection of wine spoilage	MOS			[90]
	Improving the performance of E-noses to evaluate defects	MOS	DL, SVM		[91]
	Effects of plant-derived polyphenols on the antioxidant activity and aroma of sulfur-dioxide-free red wine	MOS	PCA, LDA	GC-MS, Sensory analysis	[92]
	Artificial diagnosis of Brettanomyces spp. in Valpolicella wines	QMB	PCA	Wet chemistry	[93]
	Fast detection of TCA	MOS	PCA	Wet chemistry	[94]
	Rapid and non-destructive analysis of corky off-flavors	MOS	PCA	Wet chemistry	[95]
	Portable Electronic Nose for 2,4,6-trichloroanisole	MOS	PCA	Wet chemistry	[96]
	Detection using Electronic Nose system: application focused on spoilage thresholds by acetic acid	MOS	PCA, SVM	Wet chemistry	[97]
**Wine and grape quality**	Differentiate musts of different ripeness degree and grape varieties	MOS	PCA, PNN	Wet chemistry	[98]
**Category**	**Application**	**Sensor** **Arrays**	**Chemometrics**	**Classical**	**Reference**
**Wine and grape quality**	E-Nose applications for fruit identification, ripeness, and quality grading	CP	SAW	Wet chemistry	[99]
	characterization of flavor frame in grape wines	MOS	PCA	HS-GC-MS	[100]
	Wine quality for Shiraz vertical vintages	MOS	ML	GC-MS	[101]
	Quality assessment of wine vertical vintages	MOS	ML	GC-MS	[102]
	On-chip Electronic Nose for wine tasting Effect of leaf removal on volatile organic compound	MOS		Wet chemistry	[103]



		QMB	PCA	GC-MS	[104]
**Winemaking techniques**	Monitoring combinations of yeasts and nutrients on the aromatic profile wines GC-Electronic nose for the selection of winemaking protocol	MOS	PCA	Sensory Analysis Wet chemistry	[105]




		MOS	PCA		[106]
	Performance of a novel β-glucosidase for aroma enhancement of wines	MOS	PCA-PLSR	Sensory analysis, GC-MS	[107]
	An alternative and sustainable technique to carbonic maceration	QMB	PCA	GC-MS	[108]
	Differentiation through E-nose data modeling of rosé sparkling wines elaborated via traditional and Charmat methods	QMB	PLS-DA	GC-FID	[109]
	Early detection of postharvest noble rot in grapes	QMB	PLS-DA	Wet chemistry	[110]
**Category**	**Application**	**Sensor** **Arrays**	**Chemometrics**	**Classical**	**Reference**
**Winemaking techniques**	Influence of glutathione and ascorbic acid treatments during vinification	MOS		Wet chemistry	[111]
	Use of an Electronic Nose as a tool to differentiate winemaking techniques	MOS	ANN, PCA		[112]
**Wine and grape** **identification**	Low-cost Electronic Nose for red wine identification.	Metalloporphyrin	PCA, ELM	Wet chemistry	[113]
	Can sensory analysis and E-Noses support the assessment work	QMB	PCA	Sensory analysis	[114]
	Grape cultivar identification and classification by machine olfaction analysis	MOS	PCA, LDA, QDA, SVM, ANN	Wet chemistry	[115]
	Application of an Electronic Nose to the study of the parentage of Romanian grape varieties Identifying wine grape aromatic maturity	MOS		GC-FID	[116]




		QMB	PCR, PCA	GC-MS	[117]
**Wine and grape** **identification**	Volatile compounds in wines obtained from different managements of vineyards	MOS	PCA, LDA		[118]
	Key indicators for the discrimination of wines by E-Nose	MOS-QMB	LDA	Wet chemistry	[119]
	A multitask learning framework for multi-property detection of wine	MOS	PCA	Wet chemistry	[120]
	Effect of swelling agent treatment on grape fruit quality and the application of electronic nose identification detection	MOS	LDA, SVM	GC-MS	[121]
**Wine** **characterization**	Classification of wine faults using a low-cost electronic	MOS	ML	Wet chemistry	[122]
**Category**	**Application**	**Sensor Arrays**	**Chemometrics**	**Classical** **Comparison**	**Reference**
**Wine** **characterization**	Quantification of wine mixtures with an E-Nose	MOS	PLS, ANN	Sensory analysis	[123]
	Aroma characterization of Petit Manseng	CP	ANOVA	GC-MS	[124]
	Low-cost E-Nose for wine variety identification	MOS	PCA	Wet chemistry	[125]
	Evaluation by a GC E-Nose of the differences in volatile profile sweet wines	MOS	PCA	GC-MS	[126]
	Sniffing like a wine taster: strategy enhances E-Nose odor recognition capability	MOS	SVM		[127]
	Valuation of red wine acidification using an E-Nose system	MOS			[128]
	Authentication and composition quantitation of red wines	MOS	PCA, PLS	Wet chemistry	[129]
	Gas sensors modified with zeolite for wine aroma compounds	MOS	PCA	Wet chemistry	[130]
	E-Nose for Peruvian wine classification	MOS	PCA	GC-HPLC-MS	[131]
	Bionic electronic nose based on MOS Sensor array and machine	MOS	ML, SVM		[132]

**Table 2 sensors-24-02293-t002:** Main applications of the electronic nose for smoke taint detection.

Application	Sensor Arrays	Chemometrics	Classical Comparison	Reference
Smoke contamination in grapevine berries and taint in wines due to bushfires using a low-cost E-Nose	MOS	ANN, Sensory analysis	HPLC-MS	[149]
Smoke taint detection in pinot grigio wines using an E-Nose and machine learning algorithms	MOS	ANN		[140]
Novel digital technologies to assess smoke taint in berries and wines due to bushfires	MOS	ANN		[150]
Novel digital technologies to assess smoke taint in wine	MOS	ML, ANN		[151]
Effects of grapevine smoke exposure and technologies to assess smoke contamination and taint in grapes and wine	MOS	ML, ANN		[152]
Volatile aromatic compounds in smoke-tainted cabernet sauvignon wines using a low-cost E-Nose	MOS	ANN	GC-MS	[153]
First application of ozone postharvest fumigation to remove smoke taint from grapes	QMB	PCA, PLS		[154]

## Data Availability

Not applicable.

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
