# Peer review of "Recent Advances and Future Perspectives in the E-Nose Technologies Addressed to the Wine Industry"

_sensors, 2024, doi:10.3390/s24072293_

Round 1

Reviewer 1 Report

Comments and Suggestions for Authors

A comprehensive review of electronic nose technology is presented, with some modification suggestions as follows:

(1) References should not be cited in the conclusion.

(2) ANNs are designed for artificial neural networks, and the ANN mentioned later are not defined.

(3) There is a lack of literature review on the combination of sensory analysis of red wine and electronic nose technology.

(4) The article on adulteration of red wine should be summarized.

(5) The article on the storage period of red wine should be summarized.

(6) The relevant literature on the combination of red wine processing technology and electronic nose technology should be summarized.

Comments on the Quality of English Language

Minor editing of English language required

Author Response

Dear Referee,

please find in the attached document the answers to your comments and suggestions related to our original manuscript.

Many thanks.

Reviewer 2 Report

Comments and Suggestions for Authors

The manuscript investigates the current trends in utilizing E-nose technology for determining Volatile Organic Compounds (VOCs) and describing aromatic characteristics within the realm of oenology. To offer a comprehensive understanding of the topic, an initial examination of the state-of-the-art E-nose concept has been conducted, along with a succinct overview of recent literature spotlighting the main applications of E-nose technology in oenology. Overall, the importance of such a review paper lies in synthesizing current knowledge, identifying emerging trends, and providing guidance for researchers, practitioners, and industry professionals interested in leveraging e-nose technology for wine analysis and monitoring. I think this manuscript needs substantial revision to be accepted as a review article. Below are some comments and questions that can be effective in improving the article:

Introduction:

1-    When discussing the significance of human olfaction, it would strengthen the argument to cite specific studies or experiments that demonstrate humans' olfactory capabilities.

2-    Consider a smoother transition to explicitly introduce the relevance of E-nose technology in oenology earlier in the introduction.

3-    It would be beneficial to expand on how traditional sensory analysis methods have limitations and how E-nose technology addresses these limitations in the introduction section.

Review methodology:

4-    What criteria were used to determine the relevance of articles for discussion?

5-    Consider adding information on any tools or software used for managing references and tracking the selection process to ensure reproducibility and transparency.

Olfactory evaluation of wine: Wet chemistry and sensory analysis:

6-    Provide more specific examples or descriptions of the various classes of VOCs mentioned, such as terpenes, methoxypyrazines, and thiols.

7-    Emphasize how sensory evaluation provides insights into consumer preferences and perceptions beyond chemical composition alone.

8-    Provide a clearer comparison between sensory analysis and wet chemistry methods in terms of their strengths and limitations.

9-    Improve your review paper by adding new references in field of electronic nose for VOC detection such as DOI: 10.1016/j.snb.2022.131418 and DOI: doi.org/10.3390/s23187885.

Application of electronic noses in oenology: Principles, use and main issues:

10- What are the specific mechanisms by which different types of sensors in E-noses detect and differentiate volatile compounds in wine matrices?

11- How do environmental factors such as temperature, humidity, and atmospheric conditions affect the performance and reliability of E-noses in wine analysis?

12- Investigate the potential synergies between E-nose technology and other emerging analytical techniques such as electronic tongues or hyperspectral imaging for comprehensive wine analysis.

13- Consider a brief expression about exploring the feasibility and efficacy of miniaturized or portable E-nose devices for on-site or in-field wine quality assessment, considering practical challenges such as power consumption and data transfer.

14- Investigate the impact of wine matrix complexity, including factors such as grape variety, fermentation conditions, and aging processes, on the detection capabilities and accuracy of E-noses for different applications in oenology.

15- Discuss the implications of using E-nose technology for regulatory compliance and quality assurance in the wine industry, including considerations related to standardization, validation, and acceptance by regulatory bodies.

Case Study: E-nose in application to detect smoke taint in wines:

16- How can E-nose technology effectively detect and quantify the VPs?

17- Provide additional insights into the practical implications and potential benefits of adopting E-nose technology for early detection and mitigation of smoke taint in wines, including economic savings, quality assurance, and consumer satisfaction.

18- Discuss the challenges and opportunities associated with standardizing E-nose-based detection methods across different wine regions and production systems, considering variations in grape varieties, winemaking practices, and smoke exposure levels.

19- What are the potential limitations or challenges associated with using E-nose technology for assessing smoke contamination in wines, and how can these challenges be addressed or overcome?

20- Highlight the importance of collaborative research efforts between academia, industry, and regulatory agencies to establish consensus guidelines and best practices for implementing E-nose technology in smoke taint assessment and management in the wine industry.

Comments on the Quality of English Language

1-    Consider restructuring some sentences for improved clarity and flow. For instance, the sentence starting with "As clear" (line 44) could be rephrased for better readability.

2-    Be careful about using dots in text for example line 239.

Author Response

(The authors gave the same response as above.)

Reviewer 3 Report

Comments and Suggestions for Authors

The work is a comprehensive study of the application of the electronic nose to the wine field. The paper is clearly written. The case study is an important and interesting one.

The only issue with the paper is that there is a mismatch in many bibliographic references, so a thorough revision must be done. Only to cite an example: references 61 to 76 are related to wine aroma and wine tasting whereas in the text they are cited as composition and functioning of the different sensors

Author Response

(The authors gave the same response as above.)

Round 2

Reviewer 2 Report

Comments and Suggestions for Authors

N/A